# On Grid Quorums for Erasure Coded Data

**DOI:** 10.3390/e23020177

**Published:** 2021-01-30

**Authors:** Frédérique Oggier, Anwitaman Datta

**Affiliations:** 1Division of Mathematical Sciences, Nanyang Technological University, Singapore 639798, Singapore; frederique@ntu.edu.sg; 2School of Computer Science and Engineering, Nanyang Technological University, Singapore 639798, Singapore

**Keywords:** erasure coding, distributed storage, quorums, consistency

## Abstract

We consider the problem of designing grid quorum systems for maximum distance separable (MDS) erasure code based distributed storage systems. Quorums are used as a mechanism to maintain consistency in replication based storage systems, for which grid quorums have been shown to produce optimal load characteristics. This motivates the study of grid quorums in the context of erasure code based distributed storage systems. We show how grid quorums can be built for erasure coded data, investigate the load characteristics of these quorum systems, and demonstrate how sequential consistency is achieved even in the presence of storage node failures.

## 1. Introduction

We consider the problem of consistency for erasure code based distributed storage systems. Distributed storage systems store data over a network of nodes, so that data remains available over time. In order to do so, several properties are desirable, such as high fault-tolerance and low storage overhead. Fault-tolerance refers to the system’s ability to sustain failures of some of its components, and is present across three dimensions: (1) availability (the data should remain available even in the event of failures), (2) persistence, (the data should remain available over time), and (3) consistency (irrespective of the sequence of read and write operations on the stored data by multiple processes, and of possible failures, the data should appear to every process as if it had been manipulated in a globally agreed order).

Availability and persistence are achieved through redundancy. The data is stored multiple times, so that even when a node is unavailable, the requested data may be queried from another node (achieving (1)). Since failures may be temporary or permanent, redundancy needs to be replenished via maintenance mechanisms, in order to achieve (2), that is persistence over time. However, since the data is stored redundantly, it becomes essential to ensure consistency, so that all applications accessing a given data see the same version, in particular after updates, irrespective of which storage nodes are accessed.

Both maintenance of adequate level of redundancy and consistency depend on the redundancy mechanisms chosen, which typically induce trade-offs with storage overhead. Replication has been the most common way to ensure fault-tolerance, though over the past decade, more and more storage systems have adopted erasure coding techniques instead, e.g., Reference [1,2,3,4], since they provide a good trade-off between fault-tolerance and storage overhead. Processes to maintain the amount of redundancy over time in the presence of node failures for erasure code based distributed storage systems have been profusely studied (see, e.g., Reference [5,6] for surveys on erasure coding techniques enabling redundancy maintenance in distributed storage systems). Designs of mechanisms that support efficient updates of coded data have also been considered; see, e.g., References [7,8,9,10].

### 1.1. Consistency

In the context of replication, consistency refers to a setting where read and write operations are performed on shared data (the replicas) by different processes (see the left of Figure 1), and it informally means that, when one replica is updated by one of the processes, it should be ensured that the other copies are updated accordingly. This is achieved by fixing a set of rules the processes obey when they want to read or write the data, in exchange for which the data the processes obtain is expected to be up-to-date.

Under strict consistency (see the upper right of Figure 1), processes ask for a read operation r(*x*)*c* or a write operation w(*x*)*c* (respectively, reading or writing the value of *x* to be *c*) at a given point of the so-called wall-clock time, and the execution is expected to be instantaneous and thus follow that same ordering. The wall-clock time represents an absolute global time, while, in practice, different processes in a distributed system may not be perfectly synchronized or aware of what other processes locally consider as the time. In this example, $P_{2}$ writes w(*x*)*b* after $P_{1}$ as per the wall-clock time; thus, $P_{3}$, $P_{4}$ get *b* on every occasion when they carry out a read operation subsequently as per wall-clock time. This corresponds to the ordering w(*x*)*a*, w(*x*)*b*, r(*x*)*b*, r(*x*)*b*, r(*x*)*b*, r(*x*)*b*.

In contrast, sequential consistency only requires for some global ordering of the operations. We quote Reference [11] to provide the precise way it has been defined “… the result of any execution is the same as if the operations of all the processors were executed in some sequential order, and the operations of each individual processor appear in this sequence in the order specified by its program.” The lower right of Figure 1 illustrates an example scenario meeting this definition. The operations shown are not strictly consistent because $P_{3}$ reads *a* from *x*, even though $P_{2}$ has in real time (wall-clock time) already issued a write *b* request. Nevertheless, in this case, the following global ordering w(*x*)*a*, r(*x*)*a*, w(*x*)*b*, r(*x*)*b*, r(*x*)*b*, r(*x*)*b* specifies a legitimate sequential order of operations.

There are several other forms of consistency, including, for example causal and eventual consistencies; see, e.g., Reference [12] (Section 7): strict and sequential consistencies are strong forms of consistency, they have the advantage of maintaining a high level of global consistency at all times, at a cost in terms of latency. In this work we are interested in designing a mechanism for achieving sequential consistency over erasure coded data, and we do so by applying a standard technique to achieve so, namely quorum systems. A quorum system is defined as a collection of subsets of nodes (called quorums), where each pair of quorums has a non-empty intersection [13] (Def 3.4). A “vote" (or a “lock”) is attributed to every node in the system, and any application wishing to either read or write data needs to gather enough votes in order to perform its operation. Because of the intersecting property of quorums, mutual exclusion of a write operation with any other write or read operations is achieved.

### 1.2. Related Works

In order to achieve sequential consistency, it is necessary that multiple processes cannot carry out write operation(s) to change the value of any object, while other operations are reading or writing said value; and conversely no read operation should be carried out while a write operation is underway, i.e., mutual exclusion of any write operation from other write or read operations is needed. One way to achieve this is by means of quorum systems (we describe quorum systems in detail, later in this paper), which are subsets of nodes which intersect pair-wise. Coupled with locking mechanisms, such intersecting subsets of nodes can be used to guarantee the necessary mutual exclusion, enabling the design of protocols to enforce sequential consistency. We refer to Reference [13] for a more detailed treatment of how quorum systems are used in storage applications.

Apart from quorum systems, another popular mechanism for consistency is the class of primary-based protocols, where each data block *x* has an associated primary, which is responsible for coordinating operations on *x*. For example, Reference [9] proposed an update model assuming a primary which serializes the update executions, and, similarly, in Reference [10], the node storing a data block enforces serialized writes, while updates are disseminated in a best effort manner to the redundant blocks, and stale reads are possible, i.e., there is no consistency guarantee.

Finally the Paxos family of protocols [14] for solving consensus algorithms has been adapted to erasure coded data. For example, Reference [15] applies Paxos over erasure coded data by assuming a bound on the deviation of local clocks at nodes, without leveraging the structural properties of codes. Then, Reference [4] is an erasure code based object storage which realizes consistency using Paxos, but Paxos is used at object granularity. Coded Atomic Storage (CAS) [16] is aimed at mimicking shared memory abstraction for erasure coded data, with an emphasis on reducing communication cost, followed up by Reference [17], which builds upon Reference [16] to explore how reconfiguration of the system can be carried out while maintaining atomicity.

### 1.3. Contributions

Quorums exist in two renditions: symmetric, when a single system of sets is used to represent both reads and writes (though the kind of “vote” or “lock” associated with the acquired quorum can be different, depending on the purpose being a read or write operation), and asymmetric, when there are two distinct set systems, representing read quorums and write quorums separately. When a process uses a quorum, possibly accessing all nodes in this quorum, this induces a load, which measures the access probability of the busiest node in the system. The goal of this paper is to study a class of symmetric quorums called grid quorums [13] (Section 3.2), in the context of erasure coded data. This is motivated by the knowledge that grid quorums over replicated data exhibit optimal load characteristic and are, thus, good candidates to be generalized to the context of erasure coded data.

The contributions of this work are as follows:

(i) We specify requirements for quorum systems in the context of systematic maximally distance separable (MDS) erasure coded data (in Section 3.1). Our definition encapsulates sufficiency for the quorum system (it does not preclude the existence of different other quorum systems) to meet read/write mutual exclusion needs in the system, which in turn guarantees sequential consistency congruent to the global ordering of quorums formed.

(ii) We demonstrate (in Section 3.3) how to realize different variations $Q_{grid}^{cod1}$ and $Q_{grid}^{cod2}$ of grid quorum systems for erasure coded data subject to the quorum specification referred above. The former variant $Q_{grid}^{cod1}$ involves subsets of nodes which occupy a full row and a full column of a logical grid layout of nodes, while the latter, i.e., $Q_{grid}^{cod2}$, is a variant of the former, comprising truncated (smaller) groups still meeting the mutual intersection property. We prove that, for (n,k) MDS codes, where *n* is a square and $k=\sqrt{n}\frac{\sqrt{n}+1}{2}$, the load of the quorums under access strategy *S* are
$$L_{S}\left(Q_{grid}^{cod1}\right)=\frac{2}{\sqrt{n}},$$
and
$$L_{S}\left(Q_{grid}^{cod2}\right)=\left(2\sqrt{n}-1\right)P_{d}+P_{p}\left(\sum_{\begin{array}{c} j=1\\ j\neq \sqrt{n}-1 \end{array}}^{\sqrt{n}}\frac{\sqrt{n}-1}{\sqrt{n}-1+j}+\sum_{\begin{array}{c} j=1\\ j\neq \sqrt{n} \end{array}}^{\sqrt{n}}\frac{\sqrt{n}}{\sqrt{n}+j}\right).$$ The access strategy *S* depends on the probabilities $P_{d},P_{p}$ of accessing, respectively, data and parity nodes. Load is a metric determined by the fraction of time a given node is used, be it for a read or write operation, and is an important metric to characterize the performance and impact of a quorum system. Intuitively, lower the load, the freer the nodes in the system are to carry out other tasks. In Section 2.2 we provide a comprehensive definition for the load of a quorum.

(iii) In Section 3.4, we extend our study to *B*-grid quorums, a generalization of grid quorums. *B*-grid quorums suitable for erasure coded data are proposed which accommodate (n,k) MDS code with n=cbr and $k=n-rb^{2}$ and, thus, a wide range of rates. Their load is also computed, giving
$$L_{S}\left(Q_{B-grid}^{cod}\right)=\frac{1}{b}\left(1+\frac{1}{r}\right).$$ We then discuss the trade-offs between load and storage overhead.

(iv) In Section 4, we demonstrate how sequential consistency is achieved using the proposed quorum systems even in the presence of various combination of faults.

## 2. Background

### 2.1. Erasure Coding for Storage

A linear (n,k) erasure code over some finite field $F_{q}$ is a linear map: $\left(x_{1},\ldots ,x_{k}\right)\mapsto \left(x_{1},\ldots ,x_{k},x_{k+1},\ldots ,x_{n}\right)$, where $x_{p}$, p=k+1,…,n are linear combinations in $x_{1},\ldots ,x_{k}$, referred to as parities:(1)$$x_{p}=\sum_{i=1}^{k}a_{ji}x_{i}.$$ The vector $\left(x_{1},\ldots ,x_{k},x_{k+1},\ldots ,x_{n}\right)$ is called a codeword, and since the first coefficients of the codeword are $x_{1},\ldots ,x_{k}$, the code is said to be systematic. In a storage system, $x_{1},\ldots ,x_{k}$ are data blocks to be stored, and, in order to provide fault tolerance, the *n* coefficients $x_{1},\ldots ,x_{n}$ are stored over *n* nodes, say node *d* stores $x_{d}$, d=1,…,k for the data blocks, and node *p* stores $x_{p}$, p=k+1,…,n for the parity blocks. If the node *i* is unavailable, $x_{i}$ should be recoverable from the remaining n−1 blocks. In case several nodes are unavailable, the data content may or not be recoverable depending on the erasure tolerance ability of the code. Codes with the best fault tolerance with respect to *k* and *n* are called maximum distance separable (MDS) codes, and can tolerate a loss of up to n−k blocks. The ratio k/n is called the rate of the code. In the context of storage, people often use the reciprocal, indicating storage overhead n/k instead.

Non-MDS erasure codes with better repairability properties have been heavily researched [5,6] and even deployed in some practical systems [1]; however, as an initial study on the topic of quorum systems for erasure coded data, we consider only MDS codes, both for keeping the study relatively simpler, as well as because they continue to be widely used [2,3].

**Example** **1.**
*The (n,1) repetition code that maps $x_{1}$ to $\left(x_{1},\ldots ,x_{1}\right)$ is an MDS code, $x_{1}$ may be recovered as long as at least one out of its n copies is available. This means the storage system is keeping replicas of the data to be stored. The (n,n−1) parity check code that maps $\left(x_{1},\ldots ,x_{n-1}\right)$ to $\left(x_{1},\ldots ,x_{n-1},x_{1}+\ldots x_{n-1}\right)$ is another example. Only one erasure is tolerated. The most popular class of MDS codes is called Reed-Solomon codes. They may be defined by polynomial evaluation: coefficients of the polynomial are formed from the data to be encoded, and codewords are obtained by evaluating this polynomial.*


### 2.2. Quorum Systems

**Definition** **1.**
*Given n nodes, a quorum system Q is a set of subsets of nodes called quorums, such that every two quorums intersect, i.e., $Q\cap Q^{\prime }\neq ⌀$ for all $Q,Q^{\prime }\in Q$.*


We will label the set of *n* nodes by {1,…,n}.

**Example** **2.**
*For example, the smallest quorum system consists of just one quorum, itself consisting of one node: $Q_{sing}=\left\{\left\{i\right\}\right\}$, for some i in {1,…,n}. It is called the singleton quorum $Q_{sing}$. The majority quorum system $Q_{maj}$ is defined to be all quorums of size $\lfloor \frac{n}{2}\rfloor +1$. For example, if the set of nodes is {1,2,3,4}, then $\lfloor \frac{n}{2}\rfloor +1=3$, and $Q_{maj}=\left\{\left\{1,2,3\right\},\left\{1,2,4\right\},\left\{1,3,4\right\},\left\{2,3,4\right\}\right\}$.*


Quorums are used for maintaining consistency of data that is being read and written to, by multiple processes. In order to read or write data, and, respectively, read or write locks on, a quorum of nodes must first be obtained. If the operation is a write, a quorum Q∈Q is formed for the write operation to be performed, and, while the operation is being processed, no other operation ought to be carried out, preventing reading of stale information, or for multiple writes to overwrite over each other. This mutual exclusion is achieved using quorums and write locks, since any other operation would need to likewise acquire another quorum of nodes, which cannot be obtained since every $Q^{\prime }$ intersect with *Q* but write locks are treated to be exclusive. Multiple read operations can, however, be allowed, even with intersecting quorums, by associating read locks that are not mutually exclusive, distinct from write locks which are exclusive. Furthermore, whenever a write operation is involved, a subsequent read or write operation is guaranteed to know of the latest update because of the intersecting quorums and the mutual exclusion achieved through the locking process. Since no other operations are possible when a write operation is being carried out (because of the mutual exclusion achieved with write locks), and because any future process will necessarily include at least one node with the latest written value (from the intersection property of the quorums), the latest update will be visible to any future read or write accesses.

The load of a node depends on how often it is accessed. For every quorum Q∈Q, an access probability $P_{S}\left(Q\right)$ is defined, for *S* an access strategy. By definition, $\sum_{Q\in Q}P_{S}\left(Q\right)=1$. Then, the load $L_{S}\left(i\right)$ of node *i* using the access strategy *S* is
(2)$$L_{S}\left(i\right)=\sum_{\begin{array}{c} Q\in Q\\ i\in Q \end{array}}P_{S}\left(Q\right)$$
so that the load induced by *S* on the system is the load imposed by the busiest node:(3)$$L_{S}\left(Q\right)=\max_{i\in \{1,\ldots ,n\}}L_{S}\left(i\right).$$
The load of a quorum system Q is the minimal load, across all possible access strategies that can be used.

**Example** **3.**
*For the load of the busiest node, we have $L(Q_{sing})=1$, since for Q={{i}}, for some i in {1,…,n}, we have $L_{S}\left(i\right)=1$. For $L(Q_{maj})>1/2$, since, in a majority quorum system, all subsets of {1,…,n} of size $\lfloor \frac{n}{2}\rfloor +1$ are included, which means (we present the argument for n even) that a given node i will belong to n−1n2 quorums out of the nn2+1 quorums. Thus, a uniform access probability gives*
$$\frac{(n-1)!}{\left(\frac{n}{2}\right)!\left(\frac{n}{2}-1\right)!}\frac{\left(\frac{n}{2}+1\right)!\left(\frac{n}{2}-1\right)!}{n!}=\frac{\frac{n}{2}+1}{n}=\frac{1}{2}+\frac{1}{n}>\frac{1}{2}.$$
*Continuing Example 2, with $Q_{maj}=\left\{\left\{1,2,3\right\},\left\{1,2,4\right\},\left\{1,3,4\right\},\left\{2,3,4\right\}\right\}$, a uniform access probability means $P_{S}\left(Q\right)=\frac{1}{4}$ for every Q∈Q; thus, $L_{S}\left(1\right)=P_{S}\left(\left\{1,2,3\right\}\right)+P_{S}\left(\left\{1,2,4\right\}\right)+P_{S}\left(\left\{1,3,4\right\}\right)=\frac{3}{4}$, and, similarly, $L_{S}\left(i\right)=\frac{3}{4}$ for i=1,2,3,4.*


## 3. Grid Quorums

In the following, we consider logical grid layouts decoupled from the physical data placement of data and parity or the physical configuration of the storage nodes in the distributed system.

### 3.1. Quorum Systems and Erasure Coded Data

The definition of a quorum system (see Definition 1) considers all nodes to play the same role, since replicated data is stored (thus, every node stores the same thing). For erasure coded data, we actually have two types of nodes: those storing the actual data blocks (nodes 1,…,k) and those storing the parities (nodes k+1,…,n). When two quorums intersect, it could thus be a priori on either data nodes or parity nodes. We add the requirement that in every intersection of two quorums, there is at least one parity node. The reason is as follows: we consider each data block to be independent of each other, hence updates on one does not alter other data blocks, while it does affect parities, which, in a MDS code, are linear combinations of all data blocks. Thus, whenever any data block is updated, parities need to be updated correspondingly. The requirement that there is always some parity node in intersection of any two quorums is, thus, formally stated as:(4)$$\exists \;p\in Q\cap Q^{\prime }\;\mathrm{for}\;\mathrm{all}\;Q,Q^{\prime }\in Q\;\mathrm{and}\;p\in \left\{k+1,\ldots ,n\right\}.$$

When an update is carried out, the write lock is released only after every parity in the quorum acquired to carry out the write operation is already updated. The parities outside the quorum too need to be updated, but this is let to happen in the background. It can be carried out efficiently using a standard technique using differentials [7,8,9,10], which is outside the scope of this work.

The quorum mechanism further needs to ensure that at least one of the latest updated parities will be present in any subsequent quorum:(5)$$\exists \;p\in Q_{t}\cap Q_{t+1}\;\mathrm{for}\;\mathrm{all}\;Q_{t},Q_{t+1}\in Q\;\mathrm{and}\;p\in \left\{k+1,\ldots ,n\right\}.$$ Recall that we want to achieve sequential consistency [11] using the proposed mechanism. Here, *t* indicates a logical marker corresponding to the *t*-th snapshot view of the system. t+1 accordingly refers to the system at the conclusion of the next execution of any operation. $Q_{t}$ and $Q_{t+1}$ accordingly refers to the quorums invoked by the operations that led the system to reach the corresponding sequentially consistent states. The above invariant should hold irrespective of whether the background update propagation process is completed, in order to ensure that a process can identify the latest data and does not inadvertently obtain stale data, thus facilitating sequential consistency.

**Lemma** **1.**
*Property (Equation 5) follows from (Equation 4).*


**Proof.** Suppose quorum $Q_{t}$ has been used for an update at time *t*. Then, all parities involved in $Q_{t}$ got updated, and since any quorum $Q_{t+1}$ intersects $Q_{t}$ in such a parity *p*, property (Equation 5) follows. □

We illustrate Lemma 1 with example instances of grid quorums on the left of Figure 2. Later, in Section 3.3 we will formally describe grid quorums for coded data. We will then also demonstrate that indeed grid quorums always satisfy the necessary property (Equation 4), but for the moment, we focus on providing an example to illustrate how (Equation 5) follows from (Equation 4). The upper triangular part of the grid comprises parity blocks, data blocks are found strictly below the diagonal. Two particular instances of quorums—one comprising data block $x_{i}$ and some parity blocks, and another comprising data block $x_{j}$ and some parity blocks—are shown. These two specific instances do have one parity block (boxed for emphasis) in common, satisfying (Equation 4). The intersection of a parity across the two write quorums would ensure that any process carrying out a new write operation will become aware of the previous update, and will be able to (and need to) incorporate that update along with its own update at all the parity nodes in its own write quorum. Note that we assume that any node which has incorporated an update also stores certain meta-information, particularly the logical time-stamp of the update [18] to enable the identification of the latest version, and also stores the previous versions (or differentials) for a period of time, until a given version is propagated to all the other storage nodes in the system, before garbage collecting obsolete versions. We next provide a sketch of how this ensures sequential consistency (shown in the right panel of Figure 2 for the data block $x_{i}$).

Without loss of generality, suppose a process $P_{1}$ asks for a write of the block $x_{d}$ (w($x_{d}$)*a*, d∈{1,…,k}) and acquires a write quorum $Q_{t}$ first, while process $P_{2}$ asks for a write of the same block $x_{d}$ (w($x_{d}$)*b*) and acquires a quorum $Q_{t^{\prime }}$ for $P_{2}$ subsequently. If this second write quorum $Q_{t^{\prime }}$ is created before any of the read quorums to process the read requests from processes $P_{3}$ and $P_{4}$, then it will result in a wall-clock ordered sequence as shown in the top right quadrant of Figure 2. As an alternate scenario, it is also possible that, when the locks for $Q_{t}$ are released after $P_{1}$ completes w($x_{d}$)*a*, a different pending operation obtains a quorum before $P_{2}$ can carry out the next write operation. For instance, the first read operation by process $P_{3}$ may be carried out ahead of $P_{2}$’s write operation, leading to r($x_{d}$)(*a*) at $P_{3}$ occurring before w($x_{d}$)*b* by $P_{2}$, while all the other read operations follow this second write operation, leading to the scenario shown at the bottom right quadrant of Figure 2. Both of these scenarios satisfy sequential consistency. We used updates to the same data object $x_{d}$ to elaborate how sequential consistency is achieved, for keeping the exposition simple. However, in general, all the write operations will account for the immediately preceding write operation (Lemma 1) on any arbitrary data object and update the parities in its quorum accordingly. The immediately next update will again transitively have access to these updates. Hence, the global ordering will be determined based on the sequence in which the quorums are acquired, yielding sequential consistency. To wrap up the current example, we emphasize that though we used a particular example to demonstrate the ideas, the arguments advanced here hold for arbitrary quorum systems satisfying (Equation 4); thus, (Equation 5).

We will next provide a description of grid quorums for replicated data, before delving into the the design of grid quorums for erasure coded data.

### 3.2. Basic Grid Quorums

We recall what are basic grid quorums used over replicas.

**Definition** **2**(Reference [13] (Section 3.2))**.**
*Suppose $\sqrt{n}$ is an integer. Then, the n nodes are arranged into a square grid of edge length $\sqrt{n}$. A basic* grid quorum system *$Q_{grid}$ consists of $\sqrt{n}$ quorums, each formed by a full row and a full column of the grid, such that rows and columns are not repeated.*

In Figure 3, n=36 nodes are arranged to form a square grid of edge length $\sqrt{n}=6$. An example Q of quorum systems is $Q=\{Q_{1}=\left[1,1\right],Q_{2}=\left[2,3\right],Q_{3}=\left[3,5\right],\;$$Q_{4}=\left[4,2\right],$$Q_{5}=\left[5,6\right],Q_{6}=\left[6,4\right]\}$, where the notation [i,j] means the union of row *i* and column *j*. The quorums $Q_{2}$ and $Q_{3}$ are shown, respectively, in yellow and in blue.

In a basic grid quorum system, the size of each quorum is $2\sqrt{n}-1$ (there are $\sqrt{n}$ nodes on each row and column, including one node in their intersection), and two distinct quorums intersect exactly in two nodes. When the access strategy *S* is uniform, the probability of access is $P_{S}\left(Q\right)=\frac{1}{\sqrt{n}}$. From (Equation 2), the load of node *i* for *S* uniform is
$$L_{S}\left(i\right)=\sum_{\begin{array}{c} Q\in Q\\ i\in Q \end{array}}\frac{1}{\sqrt{n}}=\frac{\left|Q\in Q,\;i\in Q\right|}{\sqrt{n}}.$$ Since every quorum *Q* is the union of one row and one column, and every node *i* is given a unique (row, column) allocation in the grid, say *i* is at position $\left(i_{r},i_{c}\right)$, then *i* can either appear in two quorums ($\left[i_{r},j\right]$ and $\left[l,i_{c}\right]$ for some j≠l), or once (if $[i_{r},i_{c}]\in Q$). Thus,
$$L_{S}\left(i\right)=\left\{\begin{array}{cc} \frac{1}{\sqrt{n}} & if[i_{r},i_{c}]\in Q\\ \frac{2}{\sqrt{n}} & \mathrm{otherwise}. \end{array}\right.$$

This holds for every node *i* and from (Equation 3), $L_{S}\left(Q_{grid}\right)=\max_{i\in \{1,\ldots ,n\}}L_{S}\left(i\right)$, so $L_{S}\left(Q_{grid}\right)=\frac{2}{\sqrt{n}}$ for *S* uniform.

It is known (Reference [13] (Theor. 3.18)) that the minimal load of a quorum system is lower bounded by $\frac{1}{\sqrt{n}}$, and that (Reference [19] (Prop. 4.8)) the minimal load of a quorum system where every quorum has the same size *s* is given by s/n. Since $s=2\sqrt{n}-1$ for $Q_{grid}$, this gives $L\left(Q_{grid}\right)=\min_{S}L_{S}\left(Q_{grid}\right)=\frac{2\sqrt{n}-1}{n}\approx \frac{2}{\sqrt{n}}$.

In the basic grid construction, two quorums $Q_{i}=\left[i,j\right]$ and $Q_{l}=\left[l,m\right]$ always intersect in two points ([i,m] and [l,j]). It is possible to reduce the size of the intersection to one point, as follows [20]. Once the row *i* of $Q_{i}=\left[i,j\right]$ is fixed, instead of including all nodes in the *j*th column, we take instead exactly one node from each row larger than *i* (the case where every node in which its row is larger than *i*, but is in the *j*th column itself is shown in Figure 3). The difference with the previous grid quorum is that only one out of the two intersection points ([i,m] and [l,j]) is present, and the quorums are usually smaller.

### 3.3. Basic Grid Quorums for Coded Data

To obtain a basic grid quorum system for coded data, the first step we propose is to choose a (logical) grid layout that distinguishes data nodes from parity nodes: a specific such layout, where the data blocks are in the lower part of the grid, below and including the diagonal, and the parity blocks are above the diagonal, is shown in Figure 4. This layout assumes that the code maps $k=\sqrt{n}\frac{\sqrt{n}+1}{2}$ data symbols to *n* encoded ones for *n* a square, that is, the rate of the code is $\frac{1}{2}\left(1+\frac{1}{\sqrt{n}}\right)$. There are three natural ways to define quorums based on this layout, as shown in Figure 4: (i) quorums are unions of row *i* and column *i* (on the left), (ii) quorums are unions of row *i* and truncated column *i* (in the middle), (iii) quorums compromise of a node on row *i* together with truncated row *i* and truncated column *i* which include parity nodes exclusively beside a single data node. For our discussions and analysis, we will consider (i) and (iii), which are formally defined next as $Q_{grid1}$ and $Q_{grid2}$, respectively.

**Definition** **3.**
*Given an (n,k) code where $\sqrt{n}$ is an integer and $k=\sqrt{n}\frac{\sqrt{n}+1}{2}$, suppose that the n nodes are arranged into a square grid of edge length $\sqrt{n}$ such that the k data symbols are placed below and on the diagonal of the square (in positions (i,j) with i≥j), and the parities are placed above. Two basic grid quorums for coded data are given by*
$$\begin{array}{ccc} Q_{grid}^{cod1} & = & \left\{Q_{i}=\left[i,i\right],\;i=1,\ldots ,\sqrt{n}\right\}\\ Q_{grid}^{cod2} & = & \{Q_{i,j}=\left\{\left(i,j\right)\right\}\cup \left\{\left(i,c\right),\;c>i\right\}\cup \left\{\left(r,i\right),\;r<i\right\},\;i\geq j,\;i=1,\ldots ,\sqrt{n}\}. \end{array}$$


Every quorum in $Q_{grid}^{cod2}$ has size $2\sqrt{n}-1$, which is larger than that of quorums in $Q_{grid}^{cod2}$ which is $\sqrt{n}$. But then the cardinality of $Q_{grid}^{cod1}$ is $\sqrt{n}$, while that of $Q_{grid}^{cod2}$ is $k=\sqrt{n}\frac{\sqrt{n}+1}{2}$.

**Lemma** **2.**
*Both quorum system $Q_{grid}^{cod1}$ and $Q_{grid}^{cod2}$ satisfy properties (Equation 4) and (Equation 5).*


**Proof.** It is enough to prove the first property. For $Q_{grid}^{cod1}$, given *i* and *j* (i≠j), the quorums $Q_{i}=\left[i,i\right]$ and $Q_{j}=\left[j,j\right]$ intersect at (i,j) and (j,i), and the former or latter, respectively, contains a parity (above the diagonal) when i<j and j<i. For $Q_{grid}^{cod2}$, given that it differs from $Q_{grid}^{cod1}$ only on the data blocks, the parity blocks still intersect in the same manner. □

Examples are provided in Figure 4 for an (n,k) code with n=36 and k=21. Compared with a grid quorum for replicas, there are few choices for defining this quorum system given the specificities of data and parity block layout: not any choice of pairs of (row, column) works. For example, choosing $Q_{6}=\left[6,1\right]$ would not contain any parity, and, while $Q_{6}=\left[6,2\right]$ does contain a parity, it would not intersect $Q_{5}=\left[5,3\right]$ on a parity. In fact, suppose $Q_{6}=\left[6,2\right]$ is one quorum, then one cannot find any $Q_{m}=\left[m,l\right]$ for any m≠1 that would intersect $Q_{6}$ at a parity (and not repeating any column or row), since the row in $Q_{6}$ comprises only data.

We next consider the load of $Q_{grid}^{cod1}$ and $Q_{grid}^{cod2}$, for which we need to introduce an access probability.

Since we have two types of nodes, those storing parities and those storing data blocks, we consider a different access probability $P_{S}\left(\left(i,j\right)\right)$ depending on whether i>j ($P_{S}\left(\left(i,j\right)\right)=P_{p}$ for parities) or i≤j ($P_{S}\left(\left(i,j\right)\right)=P_{d}$ for data). When deciding to form a quorum for a given node, if this node belong to several quorums, then any quorum is equally likely to be called.

**Proposition** **1.**
*Given the above setting, for any choice of $P_{p}$ and $P_{d}$, the quorum access probability $P_{S}\left(Q\right)$ is uniform for $Q\in Q_{grid}^{cod1}$. Consequently, the quorum system $Q_{grid}^{cod1}$ has load*
$$L_{S}\left(Q_{grid}^{cod1}\right)=\frac{2}{\sqrt{n}}.$$


**Proof.** Given the adopted layout, row *i* of the grid contains *i* data blocks and $\sqrt{n}-i$ parities while column *j* contains j−1 parities and $\sqrt{n}-j+1$ data blocks. Therefore, the quorum $Q_{i}=\left[i,i\right]$ is formed of $\left(i+\sqrt{n}-i+1\right)-1=\left(\sqrt{n}+1\right)-1=\sqrt{n}$ data blocks (−1 accounts for the fact that row *i* and column *i* intersect on the diagonal which contains a data block, that should not be counted twice).For a node located in position (i,j), the access probability of a quorum $Q_{i}\in Q_{grid}^{cod1}$ depends on whether it is invoked, or $Q_{j}\in Q_{grid}^{cod1}$ is invoked instead. We assume both are equally likely (introducing a probability factor of $\frac{1}{2}$ for (i,j) for the terms in the computation of $P_{S}\left(Q_{i}\right)$). If j=i, $Q_{i}$ is necessarily called. Hence, we obtain:
$$\begin{array}{ccc} P_{S}\left(Q_{i}\right) & = & P_{S}\left(\left(i,i\right)\right)+\frac{1}{2}\sum_{j=1,j\neq i}^{\sqrt{n}}\left(P_{S}\left(\left(i,j\right)\right)+P_{S}\left(\left(j,i\right)\right)\right)\\ & = & P_{d}+\frac{1}{2}\left(\sqrt{n}-1\right)\left(P_{p}+P_{d}\right). \end{array}$$ For sanity check, $\sum_{i}P_{S}\left(Q_{i}\right)=\sqrt{n}P_{d}+\frac{\sqrt{n}}{2}\left(\sqrt{n}-1\right)\left(P_{p}+P_{d}\right)=kP_{d}+\left(n-k\right)P_{p}$ since $k=\frac{\sqrt{n}}{2}\left(\sqrt{n}+1\right)$ and $n-k=\frac{\sqrt{n}}{2}\left(\sqrt{n}-1\right)$. Since $P_{S}\left(Q_{i}\right)$ does not depend on $Q_{i}$, we have thus shown that
$$P_{S}\left(Q\right)=\frac{1}{\sqrt{n}}$$
since we have $\sqrt{n}$ quorums and each are equally likely.From (Equation 2), the load of node *i* for *S* uniform is
$$L_{S}\left(i\right)=\sum_{\begin{array}{c} Q\in Q\\ i\in Q \end{array}}\frac{1}{\sqrt{n}}=\frac{\left|Q\in Q,\;i\in Q\right|}{\sqrt{n}}=\frac{1}{\sqrt{n}}\;\mathrm{or}\;\frac{2}{\sqrt{n}},$$
since node *i* belongs to at most two quorums (one, when node *i* is on the diagonal). □

Recall for comparison that, for *S* uniform, $L_{S}\left(Q_{grid}\right)=\frac{2}{\sqrt{n}}\approx \frac{2\sqrt{n}-1}{n}$, and thus $L_{S}\left(Q_{grid}\right)=L_{S}\left(Q_{grid}^{cod}\right)$, we have the same load for grid quorums with replicas as with other MDS erasure coding strategies. Note that the lower bound $1/\sqrt{n}$ still holds for the the case where coded data is stored, since (i) requiring property (Equation 4) is a particular case of symmetric quorums, and (ii) setting $P_{d}=P_{p}$ reduces to $P_{S}$.

**Proposition** **2.**
*Given any choice of parity access probability $P_{p}$ and data access probability $P_{d}$, the quorum access probability $P_{S}\left(Q_{i,j}\right)$ for $Q_{i,j}\in Q_{grid}^{cod2}$ is given by*
$$P_{S}\left(Q_{i,j}\right)=P_{d}+P_{p}\left(\sum_{c=i+1}^{\sqrt{n}}\frac{1}{i+c}+\sum_{r=1}^{i-1}\frac{1}{i+r}\right).$$
*Consequently, for n≥4, the quorum system $Q_{grid}^{cod2}$ has load*
$$L_{S}\left(Q_{grid}^{cod2}\right)=\left(2\sqrt{n}-1\right)P_{d}+P_{p}\left(\sum_{\begin{array}{c} j=1\\ j\neq \sqrt{n}-1 \end{array}}^{\sqrt{n}}\frac{\sqrt{n}-1}{\sqrt{n}-1+j}+\sum_{\begin{array}{c} j=1\\ j\neq \sqrt{n} \end{array}}^{\sqrt{n}}\frac{\sqrt{n}}{\sqrt{n}+j}\right).$$


**Proof.** Since
$$Q_{grid2}^{cod}=\left\{Q_{i,j}=\left\{\left(i,j\right)\right\}\cup \left\{\left(i,c\right),\;c>i\right\}\cup \left\{\left(r,i\right),\;r<i\right\},\;i\geq j,\;i=1,\ldots ,\sqrt{n}\right\},$$
the probability of accessing $P(Q_{ij})$ is given by
$$P_{S}\left(Q_{i,j}\right)=P_{d}+P_{p}\left(\sum_{c=i+1}^{\sqrt{n}}\frac{1}{i+c}+\sum_{r=1}^{i-1}\frac{1}{i+r}\right).$$ For sanity check, $\sum_{j\leq i}P_{S}\left(Q_{i,j}\right)=kP_{d}+P_{p}\sum_{i=1}^{\sqrt{n}}i\left(\sum_{c=i+1}^{\sqrt{n}}\frac{1}{i+c}+\sum_{r=1}^{i-1}\frac{1}{i+r}\right)$, and the factor of $P_{p}$ simplifies to $\sum_{i=1}^{\sqrt{n}}i\left(\sum_{j=1}^{\sqrt{n}}\frac{1}{i+j}-\frac{1}{2i}\right)=\sum_{i\neq j}\frac{i}{i+j}$ where i,j range from 1 to $\sqrt{n}$. This sum equals to n−k because terms in the sum can be grouped into pairs of the form $\left(\frac{i}{i+j},\frac{j}{i+j}\right)$, in which the sum is 1, and there are n−k such terms.Thus, using (Equation 2), the load of node (i,j) is
$$L_{S}\left(\left(i,j\right)\right)=\sum_{\begin{array}{c} Q\in Q\\ (i,j)\in Q \end{array}}P_{S}\left(Q\right);$$
hence, for a node (i,j) storing a data block, which belongs to a single quorum, we get
$$\begin{array}{c} L_{S}\left(\left(i,j\right)\right)=P_{d}+P_{p}\left(\sum_{c=i+1}^{\sqrt{n}}\frac{1}{i+c}+\sum_{r=1}^{i-1}\frac{1}{i+r}\right)=P_{d}+P_{p}\sum_{\begin{array}{c} j=1\\ j\neq i \end{array}}^{\sqrt{n}}\frac{1}{i+j}, \end{array}$$
and the busiest of the nodes storing data is (1,1): indeed,
$$L_{S}\left(\left(i,j\right)\right)\geq L_{S}\left(\left(i+1,j\right)\right)\Leftrightarrow \sum_{\begin{array}{c} j=1\\ j\neq i \end{array}}^{\sqrt{n}}\frac{1}{i+j}\geq \sum_{\begin{array}{c} j=1\\ j\neq i+1 \end{array}}^{\sqrt{n}}\frac{1}{i+1+j}.$$ When i=1, the right-hand sum contains the terms $1/3,1/4,1/5,\ldots ,1/(1+\sqrt{n})$, while the left-hand sum contains $1/3,1/5,\ldots ,1/(2+\sqrt{n})$. Thus, the inequality reduces to $1/4\geq 1/(2+\sqrt{n})$, which holds for n≥4. When i≥2, the right-hand sum contains $1/\left(i+1\right),1/\left(i+2\right),\ldots ,1/\left(2i-1\right),1/\left(2i+1\right),\ldots ,1/\left(i+\sqrt{n}\right)$, while the left-hand sum contains $1/\left(i+2\right),1/\left(i+3\right),\ldots ,1/\left(2\left(i+1\right)-1\right),1/\left(2\left(i+1\right)+1\right),\ldots ,1/\left(i+1+\sqrt{n}\right)$. Now, the above inequality reduces to $1/\left(i+1\right)+1/2\left(i+1\right)\geq 1/2i+1/\left(i+1+\sqrt{n}\right)$, which holds term by term.If (r,c) is storing a parity block, then (r,c) belongs to r+c quorums; more precisely, it belongs to *r* quorums $Q_{r,j}$, j≤r and *c* quorums $Q_{c,j}$, j≤c. Thus,
$$\begin{array}{ccc} L_{S}\left(\left(r,c\right)\right) & = & \sum_{j=1}^{r}P_{S}\left(Q_{r,j}\right)+\sum_{j=1}^{c}P_{S}\left(Q_{c,j}\right)\\ & = & rP_{d}+rP_{p}\sum_{\begin{array}{c} j=1\\ j\neq r \end{array}}^{\sqrt{n}}\frac{1}{r+j}+cP_{d}+cP_{p}\sum_{\begin{array}{c} j=1\\ j\neq c \end{array}}^{\sqrt{n}}\frac{1}{c+j}. \end{array}$$We have that $L_{S}\left(\left(1,1\right)\right)\leq L_{S}\left(\left(r,c\right)\right)$, r≤c:
$$P_{d}+P_{p}\sum_{\begin{array}{c} j=2 \end{array}}^{\sqrt{n}}\frac{1}{1+j}\leq \left(r+c\right)P_{d}+P_{p}\left(r\sum_{\begin{array}{c} j=1\\ j\neq r \end{array}}^{\sqrt{n}}\frac{1}{r+j}+c\sum_{\begin{array}{c} j=1\\ j\neq c \end{array}}^{\sqrt{n}}\frac{1}{c+j}\right).$$ Indeed, $P_{d}\leq \left(r+c\right)P_{d}$ and
$$\sum_{\begin{array}{c} j=2 \end{array}}^{\sqrt{n}}\frac{1}{1+j}\leq r\sum_{\begin{array}{c} j=1\\ j\neq r \end{array}}^{\sqrt{n}}\frac{1}{r+j}+c\sum_{\begin{array}{c} j=1\\ j\neq c \end{array}}^{\sqrt{n}}\frac{1}{c+j}$$
because this equality is clearly true if r=1 for any choice of *c* (then the first sum on the right-hand side is the sum of the left-hand side), and we have
$$\begin{array}{ccc} r\sum_{\begin{array}{c} j=1\\ j\neq r \end{array}}^{\sqrt{n}}\frac{1}{r+j} & \leq & \left(r+1\right)\sum_{\begin{array}{c} j=1\\ j\neq r+1 \end{array}}^{\sqrt{n}}\frac{1}{r+1+j}\\ & = & r\sum_{\begin{array}{c} l=2\\ l\neq r+2 \end{array}}^{\sqrt{n}+1}\frac{1}{r+l}+\sum_{\begin{array}{c} j=1\\ j\neq r+1 \end{array}}^{\sqrt{n}}\frac{1}{r+1+j}\\ & = & r\sum_{\begin{array}{c} l=1\\ l\neq r \end{array}}^{\sqrt{n}}\frac{1}{r+l}-\frac{r}{r+1}+\frac{r}{r+\sqrt{n}+1}+\frac{r}{2r}-\frac{r}{2r+2}+\sum_{\begin{array}{c} j=1\\ j\neq r+1 \end{array}}^{\sqrt{n}}\frac{1}{r+1+j}. \end{array}$$ But, $\sum_{\begin{array}{c} j=1\\ j\neq r+1 \end{array}}^{\sqrt{n}}\frac{1}{r+1+j}\geq \frac{\sqrt{n}-1}{r+\sqrt{n}+1}$ gives
$$\frac{r+\sqrt{n}-1}{r+\sqrt{n}+1}+\frac{1}{2}-\frac{3r}{2(r+1)}\geq 0\Leftrightarrow \frac{3}{2}\geq \frac{2}{r+\sqrt{n}+1}+\frac{3r}{2(r+1)},$$
which holds. This last computation further shows that $r\sum_{\begin{array}{c} j=1\\ j\neq r \end{array}}^{\sqrt{n}}\frac{1}{r+j}$ is increasing as a function of *r*; thus, the busiest node containing a parity is obtained by maximizing both *r* and *c*; that is, $r=\sqrt{n}-1$ and $c=\sqrt{n}$. □

From the layout of the data in the grid, intuitively, one would expect the lower most parity node to be most accessed, since it will be part of all the quorums invoked to access any of the data nodes situated in the last two rows of the grid, and these rows are most numerous in terms of data nodes. The analysis above formally confirms this intuition, and provides a closed-form formula to quantify the load. Realistically, one would expect that there will be explicit access (ignoring the access of nodes because of their participation in any quorum) of data nodes much more frequently than the parity nodes, i.e., $P_{p}<<P_{d}$. In the extreme case, when $P_{p}=0$, we will have $P_{d}=\frac{1}{k}=\frac{1}{\sqrt{n}}\frac{2}{\sqrt{n}+1}$. Then, for the access strategy $S^{\prime }$ for which $P_{p}=0$ and $P_{d}$ is uniform, $L_{S^{\prime }}\left(Q_{grid}^{cod2}\right)=\left(2\sqrt{n}-1\right)P_{d}=\frac{2\sqrt{n}-1}{\sqrt{n}}\frac{2}{\sqrt{n}+1}\approx \frac{4}{\sqrt{n}}>\frac{2}{\sqrt{n}}=L_{S}\left(Q_{grid}^{cod1}\right)=L_{S}\left(Q_{grid}\right)$.

If we relax the condition of using nodes from more than one row and one column to get a quorum, say by allowing the choice of nodes in a quorum from multiple columns, akin to Reference [20] (of which, the variant discussed above is a special case), then other quorum systems can also be identified. This leads to a generalized construction, called *B*-grid, that we discuss next.

### 3.4. B-Grid Quorums for Coded Data

We recall the definition of *B*-grid quorums for replicated data.

**Definition** **4**(Reference [19] (5.2))**.**
*Suppose that n is of the form cbr and the n nodes are arranged in a rectangular grid of br rows and c columns, where rows are grouped into b bands of r rows, where band j contains rows (j−1)r+1,…,jr for j=1,…,b. Denote the intersection of column c and band j as mini-column [[j,c]]. A quorum in the*
*B*-grid system *$Q_{B-grid}$ consists of one mini-column in every band, and a representative element in each mini-column of one band. A B-grid quorum system comprises of multiple independent B-grid quorums. In particular, mini-columns and the one band from which representative elements are chosen, are independent.*

In Figure 5, we show two quorums of a *B*-grid quorum system with three bands. The same argument used to derive the load of $Q_{grid}$ may be applied here, namely, since every quorum has the same size s=br+c−1, the minimal load is (Reference [19] (Prop. 4.8)) $\frac{br+c-1}{brc}=\frac{n/c+c-1}{n}=\frac{1}{c}+\frac{c-1}{n}$. Note that, consequently, when we have a square grid, i.e., $c=br=\sqrt{n}$, the load of the *B*-grid $\approx \frac{2}{\sqrt{n}}$.

Since quorums in $Q_{B-grid}$ intersect in the mini-columns, this suggests a possible adaptation in the context of erasure coded data as follows.

**Definition** **5.**
*Given an (n,k) code, suppose that the n=cbr nodes are arranged in a rectangular grid of br rows and c columns, where rows are grouped into b bands of r rows. Let $C=\{C_{1},\ldots ,C_{b}\}$ be a fixed set of subsets of mini-columns, where $C_{i}=\left\{\left[\;\left[i,c\left(i,\beta \right)\right]\;\right],\;\beta \in \left\{1,\ldots ,b\right\}\right\}$ contains some chosen mini-columns in band i, and $|C_{i}|=b$ for i=1,…,b. Place $rb^{2}$ parities in the mini-columns specified by C, and the $k=n-rb^{2}$ data blocks elsewhere. A quorum $Q_{ij}$ in the quorum system $Q_{B-grid}^{cod}$ consists of the b mini-columns [[1,c(1,i)]],…[[b,c(b,i)]] and of a choice of c−1 elements in the band i such that there is exactly one element per mini-column. The index j of the quorum refers to the jth choice out of the r(c−1) choices to choose one element from a mini-column of r elements, for each of the c−1 columns.*


In the proposed *B*-grid quorum for coded data, each quorum comprises c−b data nodes, and br+b−1 parity nodes. Moreover, in this setup, the code rate is $\frac{n-b^{2}r}{brc}=\frac{brc-b^{2}r}{brc}=1-\frac{b}{c}$, so we need to assume c>b. This means, a quorum system for a code with arbitrarily high rate can be realized using *B*-grid.

In Figure 6, an example is shown with b=3, $C=\{C_{1},C_{2},C_{3}\}$, with $C_{1}=\{\left[\;\left[1,1\right]\;\right],\left[\;\left[1,5\right]\;\right],$[[1,7]]}, $C_{2}=\left\{\left[\;\left[2,2\right]\;\right],\left[\;\left[2,4\right]\;\right],\left[\;\left[2,6\right]\;\right]\right\}$, $C_{3}=\left\{\left[\;\left[3,3\right]\;\right],\left[\;\left[3,5\right]\;\right],\left[\;\left[3,7\right]\;\right]\right\}$. In addition, $Q_{1j}$ (in blue) contains the mini-columns [[1,5]],[[2,2]],[[3,7]] (indicated by vertical blue lines), and one choice for *j* is represented by dotted blue lines. Similarly, $Q_{2j}$ and $Q_{3j}$ are shown in yellow and pink.

**Lemma** **3.**
*The quorum system $Q_{B-grid}^{cod}$ satisfies properties (Equation 4) and (Equation 5).*


**Proof.** It is enough to prove property (Equation 4). A quorum $Q_{ij}$ must contain an element per mini-column in band *i*; therefore, it necessarily intersects the mini-columns [[i,c(i,β)]] for β=1,…,n, and thus the other quorums, and this intersection happens in parities, since by construction the parities are placed in these mini-columns. □

**Proposition** **3.**
*Considering the same setting as for $Q_{grid}^{cod}$, for any choice of $P_{p}$ and $P_{d}$, the corresponding quorum access S is uniform. Consequently, the quorum system $Q_{B-grid}^{cod}$ has load*
$$L_{S}\left(Q_{B-grid}^{cod}\right)=\frac{1}{b}\left(1+\frac{1}{r}\right).$$


**Proof.** Since the load is defined for the busiest nodes, we only consider nodes that are in mini-columns. The probability of accessing a quorum in $Q_{B-grid}^{cod}$ is
$$\frac{1}{br(c-1)}.$$ A node (say in band *j*) experiences the maximum load in the system when it is part of the mini-column for some quorum $Q_{il}$; furthermore, it will be a representative element for the mini-column for the quorum $Q_{jl^{\prime }}$. Since there are br(c−1) quorums $Q_{jl^{\prime }}$, the corresponding contribution to the load is $\frac{1}{b}$, while, the load due to the latter is $\frac{1}{r}\frac{1}{b}$, leading to a total load of $\frac{1}{b}\left(1+\frac{1}{r}\right)$. □

To compare the computed load with the square grid from Section 3.3, suppose that $br=c=\sqrt{n}$. The load can then be rewritten as $\frac{1}{b}\left(1+\frac{1}{r}\right)=\frac{1+r}{\sqrt{n}}.$ Then, if r=1, the load is indeed $\frac{2}{\sqrt{n}}$, effectively reducing the system to the basic square grid performance in terms of load (as expected). However, in terms of grid layout, this reduction only works in the case where the erasure code is replication.

Next, we compare the loads for $Q_{B-grid}$ and $Q_{B-grid}^{cod}$:$$\begin{array}{ccc} L_{S}\left(Q_{B-grid}\right) & = & \frac{1}{c}+\frac{c-1}{n}=\frac{br-1}{n}+\frac{c}{n}\\ L_{S}\left(Q_{B-grid}^{cod}\right) & = & \frac{1}{b}+\frac{1}{br}=\frac{1}{b}+\frac{c}{n} \end{array}$$
and
$$\frac{1}{b}\geq \frac{1}{b}\frac{br-1}{cr}\Leftrightarrow \frac{br-1}{cr}\leq 1\Leftrightarrow \frac{b}{c}-\frac{1}{cr}\leq 1,$$
which is always the case, since c>b. This shows that there is a cost to pay in terms of load to use erasure codes, and the relative cost is given by
$$\gamma =\frac{L_{S}\left(Q_{B-grid}^{cod}\right)}{L_{S}\left(Q_{B-grid}\right)}=\frac{c(r+1)}{br-1+c}=\frac{r+1}{\left(-1+\frac{b}{c}\right)r-\frac{1}{c}+1+r}.$$ The function γ is illustrated in Figure 7. The rate $1-\frac{b}{c}$ is shown on the *x*-axis. Small values of *c* are chosen (3 and 4); then, values of *b* smaller than *c* are considered, yielding possible rates (shown by stars). Then, for each value of *c*, several values of *r* are fixed (r=1,2,3), these choices of parameters yield six piecewise linear functions. We observe that, when *r* increases, so does the load of the coded quorum system compared to the load of the uncoded one. However, for a given *r*, we also observe that increasing *c* decreases the load.

From the view point of definition of load in a quorum system, our analysis indicates that the parity nodes are the busiest. Its practical implications, and the efficacy of *B*-grid quorum systems for erasure coded storage, need to be explored accordingly. However, we note that, for read operations, the actual work (disk I/Os) will be carried out at the nodes storing data blocks, while the parity nodes will carry out further operations beyond ‘voting’ for mutual exclusion of conflicting operations, only for write operations, where the parity values are updated. System implementation accompanied with rigorous benchmarking with realistic workloads is pending; however, since erasure coding is typically used for data that is not hot, i.e., data that is not being frequently written to, the proposed quorum mechanism looks promising for practical usage.

## 4. Consistency in Presence of Node Failures

Consider a write operation w($x_{d}$)*a* on a data object $x_{d}$, 1≤d≤k at a given time *t*. Any read or write operation involving $x_{d}$ occurring at time $t^{\prime }>t$ ought to read r($x_{d}$)*a*, assuming there has been no write operation involving $x_{d}$ in the interim. Two distinct situations arise:(1)Node *d* is available: unlike in replicated systems, there is only a single node (node *d*) which stores $x_{d}$, 1≤d≤k; thus, w($x_{d}$)*a* does update $x_{d}$ and subsequent read/write operations will obtain the updated value *a* from $x_{d}$. That some of the parity nodes may not have received all updates possibly involving other data objects has no bearings in obtaining the latest value of $x_{d}$.(2)Node *d* is unavailable: if the node storing $x_{d}$ is temporarily or permanently down, a read or write request will trigger a degraded read or write operation instead, where $x_{d}$ is obtained using other data blocks and parities, in which case, it is critical that parities involved in the degraded operation are up-to-date with respect to $x_{d}$. This scenario also needs to take into account that (i) other nodes than *d* might have been updated in the interim, leading to changes in nodes storing parities, and/or (ii) other nodes may be unavailable.

From Lemmas 2 and 3, we established that sequential consistency is achieved in the first case. We next discuss the second case, namely the consequences of node failures, and demonstrate how sequential consistency is still achieved. We adopt a level of abstraction that assumes there are mechanisms to deal with locks so that quorums are eventually secured for write and read operations, ensuring the liveliness of the system, i.e., that it does not get in a state where it cannot progress. Accordingly, we focus on the issue of safety, specifically that the sequential consistency invariant is always met.

The data redundancy is achieved using an (n,k) MDS erasure code, which has the property that one can reconstruct any missing codeword coefficient from any *k* other coefficients. When node *d* is unavailable, for 1≤d≤k, a degraded read will try to read from *k* available other nodes, at least one of them must be a parity. Parities called for degraded operations at time $t^{\prime }>t$ must be up-to-date with respect to $x_{d}$. This requirement is guaranteed if the parities used belong to the quorum used to permit the operation w($x_{d}$)*a* at time *t*. This ensures the parity nodes carry the information with respect to the latest value of $x_{d}$ without having to rely on the propagation of updates to other parity nodes, which runs as a background process. There are $\sqrt{n}-1$ parity nodes in a basic grid quorum (and br+b−1 for the B-grid) involved in any quorum.

Consider the baseline case, where all the other data objects are also available. Thus, we need only one parity to recreate the content of the unavailable node *d*. In this set-up a degraded operation is possible as long as one parity is available, which can tolerate $\sqrt{n}-2$ (br+b−2 for B-grid) unavailable parities. We also note that this parity will be up-to-date with respect to any other $x_{d^{\prime }}$, $d^{\prime }\neq d$: indeed, if this parity is not yet up-to-date, it can first be updated before using it to reconstruct $x_{d}$ since we assume that only a single data node, namely *d*, is unavailable. In this case, using k−1 data objects $x_{d^{\prime }}$, $d^{\prime }\neq d$, $1\leq d^{\prime }\leq k$ together with an up-to-date parity from the quorum of $x_{d}$ thus guarantees that the latest value of $x_{d}$ is computed as
(6)$$x_{d}=a_{pd}^{-1}\left(x_{p}-\sum_{\begin{array}{c} l=1\\ l\neq d \end{array}}^{k}a_{pl}x_{l}\right)$$
using $x_{p}=\sum_{l=1}^{k}a_{pl}x_{l}$ from (Equation 1).

Suppose now that not only node *d* is unavailable which we would like to read, but say also node $d^{\prime }$, storing the data object $x_{d^{\prime }}$ is down. We will then need more than one parity for reconstructing $x_{d}$. Since we want to access $x_{d}$ in node *d* which was last updated at time *t*, by the above discussion, we need two parities in its quorum, which are assured to be up-to-date with respect to $x_{d}$. We would ideally need these two parities to be up-to-date with respect to all other data nodes. If these are not up-to-date with respect to the data nodes other than $x_{d^{\prime }}$, since these other data nodes themselves are available, the parities can be updated to reflect the latest value of the available data nodes. Using two up-to-date parities $x_{p}$ and $x_{q}$, we get:$$\begin{array}{ccc} x_{p} & = & \sum_{\begin{array}{c} l=1\\ l\neq d,d^{\prime } \end{array}}^{k}a_{pl}x_{l}+a_{pd}x_{d}+a_{pd^{\prime }}x_{d^{\prime }}\\ x_{q} & = & \sum_{\begin{array}{c} l=1\\ l\neq d,d^{\prime } \end{array}}^{k}a_{ql}x_{l}+a_{qd}x_{d}+a_{qd^{\prime }}x_{d^{\prime }}, \end{array}$$
so we multiply the first equation by $a_{qd^{\prime }}$ and the second by $a_{pd^{\prime }}$ and compute the difference of the two terms, which yields
$$a_{qd^{\prime }}x_{p}-a_{pd^{\prime }}x_{q}=\sum_{\begin{array}{c} l=1\\ l\neq d,d^{\prime } \end{array}}^{k}\alpha_{l}x_{l}+\left(a_{qd^{\prime }}a_{pd}-a_{pd^{\prime }}a_{qd}\right)x_{d}$$
for $\alpha_{l}$ the corresponding coefficients. Then, $x_{d}$ is found from this equation since we know $x_{p},x_{q}$ and $x_{l}$ for $l\neq d,d^{\prime }$, and only $x_{d},x_{d}^{\prime }$ are assumed unavailable. Specifically, we have
(7)$$x_{d}={\left(a_{qd^{\prime }}a_{pd}-a_{pd^{\prime }}a_{qd}\right)}^{-1}\left(a_{qd^{\prime }}x_{p}-a_{pd^{\prime }}x_{q}-\sum_{\begin{array}{c} l=1\\ l\neq d,d^{\prime } \end{array}}^{k}\alpha_{l}x_{l}\right).$$

Notice that the above computation of $x_{d}$ only assumes that $x_{d^{\prime }}$ has the same value in $x_{p}$ and $x_{q}$. Therefore, even if the data node $x_{d^{\prime }}$ needed for the update is down, it is actually enough to find two parity nodes which reflect the same (possibly stale) value of $x_{d^{\prime }}$, for being able to perform a degraded read and recreate the latest value of $x_{d}$ correctly.

Finally, it is possible that a different data node $x_{d^{\prime }}$ has been updated subsequent to the last update to $x_{d}$, but in the interim, the unique parity node (We consider the case where the data nodes are from different rows in the grid/B-grid layout, such that they have distinct set of parities in their quorums, with a single intersecting parity. If they have multiple common parities in the intersection of their quorums, then the considered problem does not arise, since all these parities in the intersection would carry information regarding the latest values for both $x_{d}$ and $x_{d^{\prime }}$.) at the intersection of the quorums required to read or write $x_{d}$ and $x_{d^{\prime }}$ becomes unavailable. Again, as above, if we use any two parities from the quorum for $x_{d}$ which reflect the same (latest or stale) value of $x_{d^{\prime }}$, and exclude $x_{d^{\prime }}$ irrespective of whether it is available or not, then we can reconstruct the latest value of $x_{d}$ in the same manner, i.e., using (Equation 7), as the above scenario.

Note that, while we have not explicitly discussed a truncated B-grid, similar to the truncated version of basic grid, where quorums were formed comprising only single data object and the parity nodes determined based on the row the data object belonged to, one can also truncate the B-grid quorums. Doing so will not alter the fault tolerance or consistency discussed above.

## 5. Concluding Remarks

This study is the first of its kind, in synthesizing the concept of quorum systems with erasure coded storage systems and showing the feasibility of grid quorums in that context. It creates a stepping stone for further studies on the fault-tolerance and availability of the proposed quorum systems, including: (1) access of the quorums for repair operations and degraded read operations, (2) new quorum systems for erasure coded systems with better characteristics, in terms of practical requirements, such as load, coding rate, and fault-tolerance, and (3) taking into account the asymmetric role of data and parity nodes to possibly define quorum load in a more meaningful manner. The design of practical algorithms, considering system design issues, including background update propagation, mapping the logical layout to the physical layout, replacement of permanently down nodes, garbage collection of stale information, and the overlying file system, are numerous aspects which will also need attention once the conceptual foundations mature, to translate the ideas into a working system.

Furthermore, in the context of storage systems, non-MDS codes but with better repairability properties have been proposed [5] and are also deployed [1] in real-world systems, where (some) parities have certain locality properties, such that they do not depend on all the data blocks. As such, the constraint from (Equation 4) may not be adequate when such codes are used, and further studies are needed. For example, it would be interesting to consider the joint design of codes with good repairability and efficient quorum mechanisms.

## Figures and Tables

**Figure 1 entropy-23-00177-f001:**
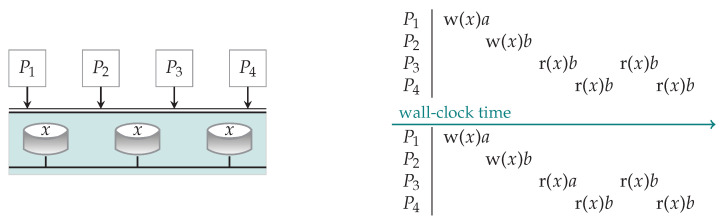
On the left: A simplified view of a distributed storage system, where nodes store replicas of some data object *x*. Processes $P_{1},P_{2},P_{3},P_{4}$ may ask to read the current value, say *c*, from *x*, represented as r(*x*)*c* or write the value *c* to *x*, represented as w(*x*)*c*. They may carry out the operations at any of the replicas. On the right, two forms of consistency are illustrated: a read operation r(*x*)*c* or a write operation w(*x*)*c* at a given point of the wall-clock time indicates that a process is asking for the corresponding operation on a replica. When it is effectively executed is inferred from the table: under **strict consistency** (illustrated in the **upper right quadrant**), the executions follow the same timeline, while under **sequential consistency** (illustrated in the **lower right quadrant**), the executions follow some global ordering but need not adhere to the wall clock time at which the operations were invoked by the processes.

**Figure 2 entropy-23-00177-f002:**
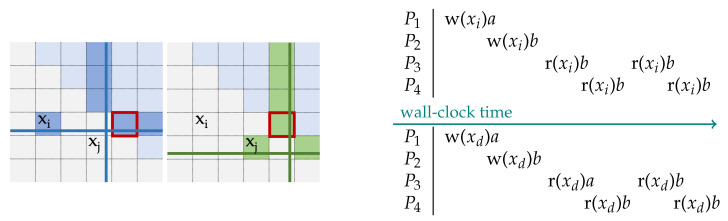
Update sequence (for the grid-quorums on the left of the figure): at time *t*, $x_{i}$ is updated using its quorum $Q_{t}$, in which parities are highlighted in row and column 4 (shown on the left). At time $t^{\prime }>t$, $x_{j}$ is updated using its quorum $Q_{t^{\prime }}$, in which parities are highlighted in row 5 and column 5 (on the right). Since the parity node (boxed for highlighting) at coordinate (4,5) is common, process updating $x_{j}$ would know that $x_{i}$ was updated, and the latest value of $x_{i}$ will have to be taken into account during its own update, so that all the parities in $Q_{t^{\prime }}$ reflect not only the latest value of $x_{j}$, but also the latest value of $x_{i}$, irrespective of whether they had received the update information regarding $x_{i}$ through the background process prior to the invocation of $Q_{t^{\prime }}$. Two distinct but valid scenarios of sequential consistency are shown (on the right of the figure) based on different sequences of quorums acquired.

**Figure 3 entropy-23-00177-f003:**
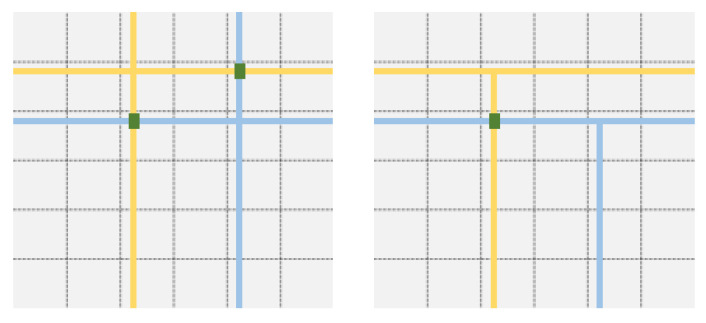
Basic grid quorums for n=36 nodes: on the left, two quorums (one in yellow and the other in blue) intersect in two points, while, on the right, a variant with smaller quorums is shown, where they intersect in one point.

**Figure 4 entropy-23-00177-f004:**
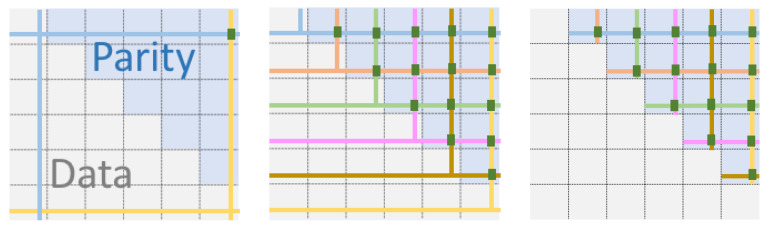
Grid (6×6) layout for an (36,21) code: Data blocks are in the lower triangle (including the diagonal), while the upper triangle has the parities. On the left, an example of two intersecting quorums $Q_{1}=\left[1,1\right],Q_{2}=\left[6,6\right]$ is shown, and they interest in two points, including a parity. In the middle, all the quorums for a variant with smaller quorums (obtained by truncating columns) are shown. On the right, each data block on row *i* has for quorum the union of itself, the parities on row *i* and the parities on column *i*.

**Figure 5 entropy-23-00177-f005:**
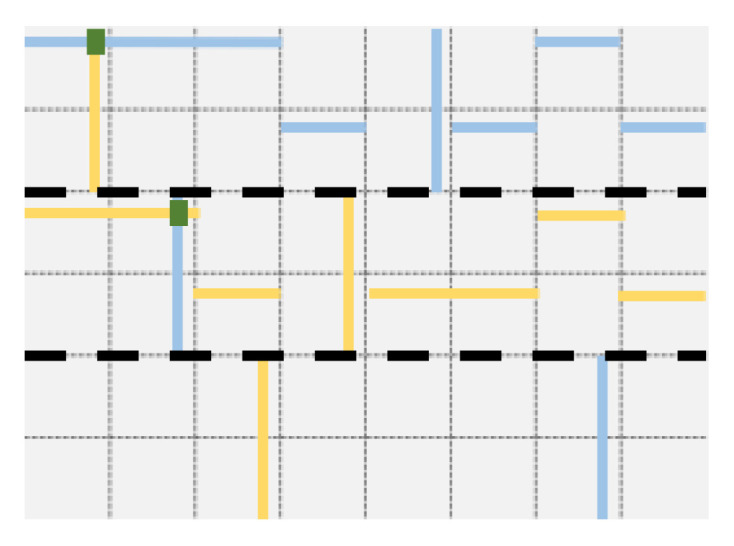
A *B*-grid quorum for n=48=cbr nodes, arranged in a grid with c=8 columns, br=6 rows, arranged in b=3 bars each containing r=2 rows.

**Figure 6 entropy-23-00177-f006:**
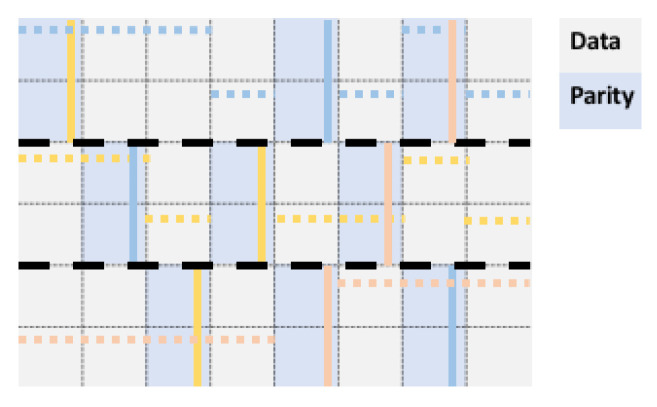
A *B*-grid with (48,30) coded data.

**Figure 7 entropy-23-00177-f007:**
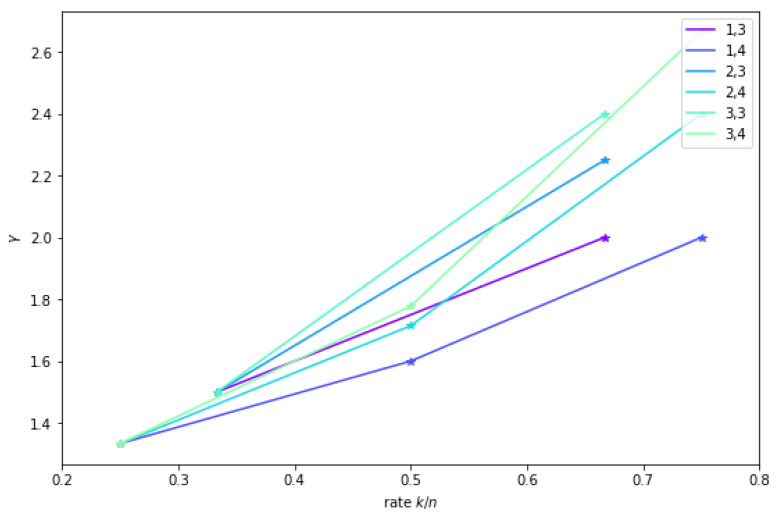
On the *x*-axis, the rate $1-\frac{b}{c}$. On the *y*-axis, the relative load γ of the coding based B-grid w.r.to replication as a function of (r,c), for (r,c)∈{(1,3),(1,4),(2,3),(2,4),(3,3),(3,4)}.

## Data Availability

Data sharing not applicable.

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
