# Peer review of "On Grid Quorums for Erasure Coded Data"

_entropy, 2021, doi:10.3390/e23020177_

Round 1

Reviewer 1 Report

This paper studies quorum systems in a coded distributed storage systems. It extends the problem formulation for uncoded distributed storage systems. When the data are in distributed in uncoded form, the data replicated in every storage nodes. A quorum system in the uncoded case is a collection of sets such that every pair of sets have nonempty intersection. 

When the data are protected by systematic erasure code, the authors propose that we need to treat the systematic nodes and the parity-check nodes separately. The problem of consistency is very different from the uncoded case. A special grid topology of storage nodes is investigated. The number of parity-check nodes is roughly half of the total number of nodes. A particular quorum system is studied. The load of the storage nodes are derived. This paper formulates the consistency problem with storage nodes failure. 

The main contribution of the paper is the re-formulation of the quorum system for erasure-coded storage system. Although the main theorem in this paper only applies to a specific example, this example illustrates some aspect of the new research problem. The paper in general is well written. I recommend acceptance for publication.

Minor comments:

In p.9 line 1, there is some typing error in the first sentence "/ADwhich comprise of parity nodes ...."

Author Response

This paper studies quorum systems in a coded distributed storage systems. It extends the problem formulation for uncoded distributed storage systems. When the data are in distributed in uncoded form, the data replicated in every storage nodes. A quorum system in the uncoded case is a collection of sets such that every pair of sets have nonempty intersection.

When the data are protected by systematic erasure code, the authors propose that we need to treat the systematic nodes and the parity-check nodes separately. The problem of consistency is very different from the uncoded case. A special grid topology of storage nodes is investigated. The number of parity-check nodes is roughly half of the total number of nodes. A particular quorum system is studied. The load of the storage nodes are derived. This paper formulates the consistency problem with storage nodes failure.

The main contribution of the paper is the re-formulation of the quorum system for erasure-coded storage system. Although the main theorem in this paper only applies to a specific example, this example illustrates some aspect of the new research problem. The paper in general is well written. I recommend acceptance for publication.

Respones: That aptly summarizes our work. We are happy to learn that the reviewer liked the work and recommends acceptance for publication.

Minor comments:

In p.9 line 1, there is some typing error in the first sentence "/ADwhich comprise of parity nodes ...."

Respones: That was indeed a typing error. Thanks for pointing it out. We have rectified it.

Reviewer 2 Report

The authors considered the application of coding theory to consistency issues in distributed computing systems, which is recent and interesting. A detailed analysis is provided for certain classes of well known codes in the framework of quorum systems, one tool set for consistency.

Some specific comments:

  1. Two notions of consistency are highlighted. While strict consistency appears easy to understand, sequential consistency is not as some violations are allowed (but what is allowed really?). From Section 1.1, I am not able to tell the exact definition of sequential consistency. Is it that later read can only be the first previous write but not second previous write etc? Maybe math will be more clear here.
  2. The connection between sequential consistency and quorum systems is not explained in Section 1.1. (and other places) thus not clear. This is important as the remaining part of the work only deals with quorum system. This might come from the fact that I do not fully understand sequential consistency as discussed in the above issue. Will be good to make the connection explicit, e.g., what kind of quorum systems will guarantee sequential consistency, and further, any quantitive relation in certain metrics?
  3. The contributions in Section 1.3 are not possible to understand as many definitions are not yet defined, e.g., the various specific quorum systems, load, probabilities, so maybe need to be rephrased.
  4. Definition 1 of quorum system appears to be weak, as only intersection is required. Could be more interesting if some joint code and quorum design is presented instead of a mere analysis (of course this could be future work).

Author Response

The authors considered the application of coding theory to consistency issues in distributed computing systems, which is recent and interesting. A detailed analysis is provided for certain classes of well known codes in the framework of quorum systems, one tool set for consistency.

Responses: We are happy to learn that the reviewer found our work interesting and timely. Thanks.

Some specific comments:

  1. Two notions of consistency are highlighted. While strict consistency appears easy to understand, sequential consistency is not as some violations are allowed (but what is allowed really?). From Section 1.1, I am not able to tell the exact definition of sequential consistency. Is it that later read can only be the first previous write but not second previous write etc? Maybe math will be more clear here.

Responses: We have added further explanation on sequential consistency, but we did so, erring on the side of caution and tradition. Specifically, the notion of sequential consistency was proposed by Lamport in 1979, yet we could locate only verbal rather than a formal mathematical definition of the term in mainstream and prominent publications on the topic. As such, we have preferred to keep the description verbal as well. What we have done now is added more explanation, including verbatim quote from Lamport, which is what most prominent works in the literature do. We believe that the expanded explanation and the canonical definition widely used in the literature addresses the concern of clarity.

  1. The connection between sequential consistency and quorum systems is not explained in Section 1.1. (and other places) thus not clear. This is important as the remaining part of the work only deals with quorum system. This might come from the fact that I do not fully understand sequential consistency as discussed in the above issue. Will be good to make the connection explicit, e.g., what kind of quorum systems will guarantee sequential consistency, and further, any quantitive relation in certain metrics?

Responses: We have added a paragraph at the start of section 1.2 to connect the ideas. We avoided doing this in section 1.1 where we summarize the notions of consistency, and instead we used section 1.2 to do so, since in that subsection we also discuss other approaches than quorums with which consistency can be achieved. As to the question of  `what kind of quorum systems will achieve sequential consistency’, the basic definition of quorums (intersecting subsets) suffices to guarantee it (subject to other elements of the protocol design) for replicated systems. We demonstrate how this can also be exploited to achieve the same even for erasure coded system; though in the case of erasure coded systems, the nature of redundancy imposes further constraints on how these subsets are to be chosen.

Responses: The other quantitative metrics (e.g., load) do not determine the functional correctness of the quorums, but instead have implications in terms of their performance. E.g., a trivial solution for consistency will be to define a quorum which requires all nodes to be involved for all reads and all write operations. However that will then induce more (and unnecessary for the purpose of consistency) load at every node. Such a mechanism will also have lower availability (the moment there is any failure, no quorums can be formed). That will inhibit progress and availability, but not hamper consistency in the system.

As such, we do characterize the properties of the quorums we have designed, but the difference in choices is not regarding the basic requirement of guarantee of consistency, their implications lie in the performance of the system.

So in brief, there is no immediate connection with the quantitative metrics, that is why we do not delve in that kind of analysis, and the reviewer did not miss anything.

  1. The contributions in Section 1.3 are not possible to understand as many definitions are not yet defined, e.g., the various specific quorum systems, load, probabilities, so maybe need to be rephrased.

Responses: We have added more details in explaining the contributions, while trying to strike a balance. Namely, at this stage, we hope that the additional detail helps a reader unfamiliar with quorum systems to understand the intuition of the ideas, but we cannot go in complete detail of the concepts and implications (which arrive at appropriate juncture of the text), and yet, we did not want to compromise on the specificities, so that a reader familiar with quorum systems can use this section to immediately get the big picture synopsis of the precise contributions.  

  1. Definition 1 of quorum system appears to be weak, as only intersection is required. Could be more interesting if some joint code and quorum design is presented instead of a mere analysis (of course this could be future work).

Responses: From a systems point of view, that is also the strength of our work. A simple design which has powerful implications. It has been so for replicated systems for decades. Our contribution is to demonstrate how to achieve that in the context of erasure coded system, without complicating the requirements too much (disclaimer: there is some additional complexity in our design, in which subsets satisfying those conditions take into account the nuance of the different kind of redundancy involved with coding).

However, we also agree, that there is room for further exploration, designing mechanisms which are more closely coupled in the design of the codes themselves along with the accompanying consistency enforcing quorums. We indicated these for future work, and appreciate that the reviewer understands that the current work is a starting point, leading way for the future exploration of more ambitious and novel approaches.